# BOOSTING NETWORK: LEARN BY GROWING FILTERS AND LAYERS VIA SPLITLBI

## ABSTRACT

Network structures are important to learning good representations of many tasks in computer vision and machine learning communities. These structures are either manually designed, or searched by Neural Architecture Search (NAS) in previous works, which however requires either expert-level efforts, or prohibitive computational cost. In practice, it is desirable to efficiently and simultaneously learn both the structures and parameters of a network from arbitrary classes with budgeted computational cost. We identify it as a new learning paradigm – Boosting Network, where one starts from simple models, delving into complex trained models progressively. In this paper, by virtue of an iterative sparse regularization path -- Split Linearized Bregman Iteration (SplitLBI), we propose a simple yet effective boosting network method that can simultaneously grow and train a network by progressively adding both convolutional filters and layers. Extensive experiments with VGG and ResNets validate the effectiveness of our proposed algorithms.

## 1 INTRODUCTION

In recent years, deep convolution neural networks have made remarkable achievements in compute vision and machine learning communities in addressing many important tasks, such as image classification, and segmentation. Researchers had designed many successful Deep Neural Network (DNN) architectures, from just have a few convolution layers like LeNet (LeCun et al., 1998) and AlexNet (Krizhevsky et al., 2012), to have more than 10 layers, *e.g.*, VGG (Simonyan & Zisserman, 2014) and GoogleLeNet (Szegedy et al., 2015), and even have hundreds and thousands of layers like ResNet (He et al., 2016a). Designing a neural network architecture requires expert-level efforts to specify the key network hyper-parameters, such as type of layers, number of filters and layers (*i.e.*, network width and depth) and so on. Since the capacity of over-parameterized networks largely depends on the number of total parameters, the number of filters and layers of networks are the key hyper-parameters that shape the expressive power of neural networks.

In machine learning communities, most researchers resort to AutoML methods, *e.g.*, Neural Architecture Search (NAS), in automating architecture engineering. Critically, NAS methods indeed surpass the manually designed architectures on many tasks, such as image classification and object detection (Zoph & Le, 2016; Zoph et al., 2018). To search a good architecture, various search strategies have been employed, such as random search, Bayesian optimization, reinforcement learning, and so on. Most of them require significant amount of computational cost, which is normally orders of magnitude higher than training a network. Furthermore, some of the found architectures by NAS have much more parameters than manually designed ones on the same dataset.

As the field of representation learning moves closer towards artificial intelligence, it becomes important to efficiently and simultaneously learn both the structures and parameters of a network from arbitrary classes on mobile devices or even Internet of Things (IoT) devices. This requires more flexible strategies in dynamically handling the network width and depth, according to the scale of dataset. To this end, this paper studies a new paradigm – *Boosting network* (BoN), where *one starts from simple models, delving into complex trained models progressively.* Specifically, BoN could simultaneously grow the structures and train the parameters from a simple initialized network on the data gradually to complex ones. Formally, we demand the following properties of an algorithm qualified as BoN:

- It should incorporate both architecture growth (including filters and layers) and parameter learning simultaneously, in which the width and depth of network can be gradually updated, and the parameters of network should be updated at the same time;

- It should provide a comparable classifier for prediction tasks, as the state-of-the-art hand-crafted architectures on the same dataset;

- Its computational requirements, the total parameters of final boosted network, and memory footprint should remain bounded, ideally in the same order of magnitude as training a manually engineered architecture on the same dataset.

The first two criteria express the essence of boosting network; the third criterion identifies the key difference from NAS and other trivial or brute-force solutions, such as randomly searching.

This paper proposes a method for the BoN task based on the Split Linearized Bregman Iteration (SplitLBI) (Huang et al., 2016; Fu et al., 2019), originally proposed by Huang et al. (2016) to learn high dimensional sparse linear models and found applications in medical image classification (Sun et al., 2017), computer vision (Zhao et al., 2018), and training neural networks (Fu et al., 2019). Particularly, based on differential inclusions of inverse scale spaces (Huang et al., 2018), SplitLBI has the merit of learning both an over-parameterized model weight set (Over-Par set) as the Stochastic Gradient Descent (SGD), and structural sparsity model weight set (Stru-Spa set) in a coupled inverse scale space. Essentially, SplitLBI optimizes the Stru-Spa set as sparse approximation of the Over-Par set, by gradually selecting the important filters and weights from Over-Par set, along the training epochs.

Equipped with SplitLBI, our key idea of BoN comes from progressively growing networks by checking the parameters within Over-Par and Stru-Spa set. Essentially along the training epochs, if enough parameters in Over-Par set have been selected in Stru-Spa set, it would be more advisable to increase the capacity of Over-Par Set by adding new parameters.

Formally, to boosting a network, we introduce a Growing and Training Network Algorithm (GT-Net Alg), consisting of two parts of growing both filters and layers, i.e., Growing and Training filters algorithm (GT-filters Alg), and Growing and Training layers algorithm (GT-layers Alg). Given an initial network, the GT-filters Alg can effectively grow the filters of each layer, and train the network parameters at the same time. Furthermore, the GT-layers Alg firstly employs GT-filters Alg to compute the filter configuration for the layers of each block, and then periodically check whether to add new layer to the block along the training procedure. We conduct extensive experiments on several benchmark datasets, including MNIST, Cifar-10, and Cifar-100. It shows that our GT-Net Alg can achieve comparable or even better performance than the competitors, with much less computational cost, and smaller size of found network. This indicates the effectiveness of our proposed algorithms. Up to our knowledge, this is the first time that a BoN type algorithm of all the three aspects above is addressed in literature.

**Contributions**. We summarize the contributions: (1) A novel learning paradigm - Boosting network (BoN), is for the first time, studied in this paper: one starts from simple models, delving into complex trained models progressively. (2) We propose a novel GT-filters Alg, which can simultaneously effectively grows the filters of each layer, and trains the network parameters, given a simple initial network. (3) We present GT-layers Alg which grows and trains layers, by exploring the over-parameterized model weight and structural sparsity model weight set.

## 2 RELATED WORK

To explore a good deep learning structure, recent research focuses on employing Network Architecture Search (NAS) (Elsken et al., 2018; Zoph & Le, 2016; Zoph et al., 2018) by using reinforcement learning to search the network structures, such as number of filters, filter size, layer depth, and so on. Despite promising performance achieved, the computational cost of NAS algorithms themselves are prohibitive expensive, *e.g.*, 800 GPUs concurrently at any time training the algorithms in Zoph & Le (2016). Several approached improved NAS by accelerating it, including weight sharing/inheritance methods , or decreasing the searching space to a specific setting (Elsken et al., 2017; Pham et al., 2018; Cai et al., 2018c; Bender et al., 2018). But they still require significant amount of computational cost.( Nonparametric Network (Philipp & Carbonell, 2017) uses group lasso to update the

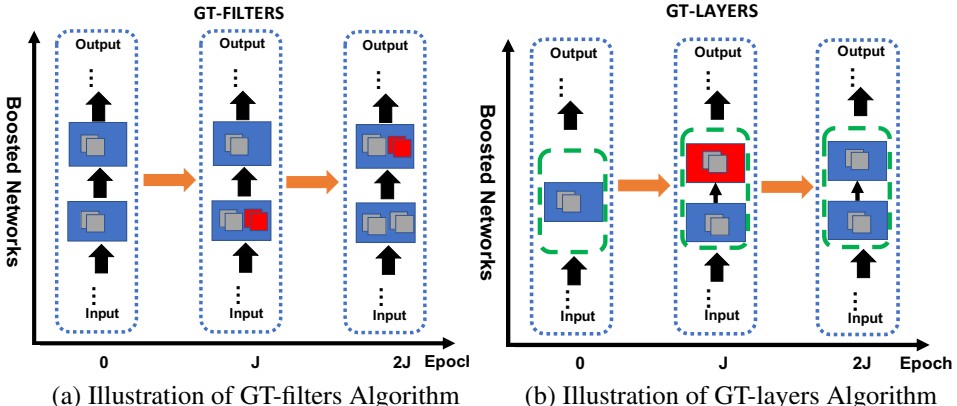

(a) Illustration of GT-filters Algorithm          (b) Illustration of GT-layers Algorithm

Figure 1: Illustration of GT-Net Algorithm. The red squares indicate the newly added filters of each layer in (a), and newly added layers of each block in (b), respectively.

network structure.) In contrast, Our BoN aims at growing a network and making a balance between computational cost, the model size and performance of the network.

Network pruning algorithms (Han et al., 2015; Abbasi-Asl & Yu, 2017; Molchanov et al., 2017) introduce additional computational cost in fine-tuning/updating the networks. In addition, some works study the manually designed compact and lightweight small DNNs (*e.g.* ShuffleNet (Ma et al., 2018), MobileNet (Howard et al., 2017), and SqueezeNet (Iandola et al., 2017)), which may still be tailored only for some specific tasks rather than boosting a network in this work.

Several recent works also consider adding layers to networks. Network Morphism (Chen et al., 2015; Wei et al., 2016; 2017; Cai et al., 2018a;b) aims at accelerating the deep networks by adding layers to a shallower net while preserving the parameters of the shallower net. One recent arxiv paper – Autogrow (Wen et al., 2019) also explores adding new layers in an automatic way. But none of these works can dynamically add filters to an existing layer as our GT-filters Alg. Additionally, these methods still requires significant computational cost and training resources.

## 3 METHODOLOGY

### 3.1 BACKGROUND: SPLIT LINEARIZED BREGMAN ITERATIONS (SPLITLBI)

Our whole algorithm is built upon the SplitLBI algorithm. The basic spirit of this algorithm (Huang et al., 2018) lies in two coupled spaces: weight parameter $W$ (Over-Par set) to explore over-parameterized models by gradient descent and structural sparsity parameter $\Gamma$ (Stru-Spa set) to explore important subnetwork architectures by inverse scale space where those important parameters become nonzero faster than others. The SplitLBI algorithm can be described as following,

$$W^{t+1} = W^t - \kappa\alpha\nabla_W L\left(W^t, \Gamma^t\right) \tag{1}$$

$$Z^{t+1} = Z^t - \alpha\nabla_\Gamma L\left(W^t, \Gamma^t\right) \tag{2}$$

$$\Gamma^{t+1} = \kappa \cdot \text{Prox}\left(Z^{t+1}\right) \tag{3}$$

where $Z^0 = \Gamma^0 = 0$; and $L\left(W^t, \Gamma^t\right) = L_{task}\left(W^t\right) + \frac{1}{2\nu}\left\|W^t - \Gamma^t\right\|_2^2$ indicates the loss function at $t$, with task specific loss $L_{task}\left(W^t\right)$ (*e.g.*, cross-entropy loss). $W$ is initialized as He et al. (2015); $\Gamma$ is learned to approximate $W$ here. And the Prox is the proximal mapping function with the following form:

$$\text{Prox}\left(Z\right) = \text{argmin}_\Gamma \frac{1}{2}\|\Gamma - Z\|_2^2 + \|Z\|_{1,2}$$

$$= \min\left(0, 1 - \frac{1}{\|Z\|_{1,2}}\right)Z \tag{4}$$

where $\|Z\|_{1,2}$ is a group Lasso ($\ell_1 - \ell_2$) norm for convolutional filters or simply the Lasso ($\ell_1$) norm for weights). The hyper-parameters of SplitLBI are $\alpha$ is the learning rate; $\kappa$ and $\nu$ are controlling the sparsity of learned model.

As in Huang et al. (2016), and assume $P := (W, \Gamma)$, the SplitLBI in Eq. (1-3) can be rewritten as standard Linearized Bregman Iteration, resulting the loss function,

$$P_{k+1} = \arg\min_{P} \left\{ \langle P - P_k, \alpha\nabla\bar{\mathcal{L}}(P_k) \rangle + B_{\Psi}^{p_k}(P, P_k) \right\}, \tag{5}$$

where

$$\Psi(P) = \Omega_{\lambda}(\Gamma) + \frac{1}{2\kappa}\|P\|_2^2 = \Omega_{\lambda}(\Gamma) + \frac{1}{2\kappa}\|W\|_2^2 + \frac{1}{2\kappa}\|\Gamma\|_2^2, \tag{6}$$

$p_k \in \partial\Psi(P_k)$, and $B_{\Psi}^{q}$ is the Bregman divergence associated with convex function $\Psi$, defined by

$$B_{\Psi}^{q}(P, Q) := \Psi(P) - \Psi(Q) - \langle q, P - Q \rangle, \text{ for some } q \in \partial\Psi(Q). \tag{7}$$

One can see that Eq. (1) is essentially an gradient descent step over the primal parameter $W_t$. However in Eq. (2-3), SplitLBI lifts the original network parameters $W$, to a coupled parameter set, $(W, \Gamma)$, where a sparse proximal gradient descent (or Linearized Bregman Iteration, Mirror Descent) runs over the dual parameter $\Gamma$ which enforces structural sparsity on network models. Along the training path, $\Gamma$ set is to learn a sparse approximation of parameter set $W$; and the important filters and parameters will gradually become non-zeros in $\Gamma$.

## 3.2 GROWING AND TRAINING FILTERS ALGORITHM (GT-FILTERS ALG)

Built upon the SplitLBI, we further propose a GT-filters Algorithm of learning to expand the conv filters and train network parameters simultaneously. Specifically, starting from very few filters of each conv layer, GT-filter algorithm requires not only efficiently optimizing the parameters of filters, but also adding more filters if the existing filters do not have enough capacity to model the distribution of training data.

Remarkably, our boosting network is very different from previous tasks, including AutoML (Wong et al., 2018), or life-long learning (Wang et al., 2017; Thrun & Mitchell, 1995; Pentina & Lampert, 2015; Li & Hoiem, 2016), knowledge distill (Hinton et al., 2014). In general, these existing works do not allow additional expanding or fine-tuning algorithms which are very computational expensive in practice. In our GT-filters Alg, we define a projection of the conv filters to grow and train filters, as the $W^t$ onto the support set of $\Gamma^t$,

$$\widetilde{W}^t = \text{Proj}_{\text{supp}(\Gamma^t)}\left(W^t\right). \tag{8}$$

This above equation means $W^t$ is projected on $\Gamma^t$, and the selected subset $\widetilde{W}^t$ include the parameters existed in both $W^t$ and $\Gamma^t$. The basic idea is to monitor the gap between $W^t$ and its projection $\widetilde{W}^t$ along the training iterations: when the gap becomes small, we are going to expand the network by adding new filters.

Fundamentally, the expressive power of recent deep convolutional neural networks largely attributes to the model over-parameterization. As in Eq. (1), the parameter set $\Gamma$ sparsely approximate the weight set $W$. Thus intuitively, we can employ Eq. (8) to indicate whether the network is over-parameterized: if the set of $\widetilde{W}^t$ is much smaller than that of $W^t$, that means the model is well over-parameterized and of enough capacity for the task at current iteration step; otherwise, if we have,

$$|\widetilde{W}^t|/|W^t| > \tau, \tag{9}$$

then it would be more advisable to enlarge the model capacity by adding filters. Here $\left|\widetilde{W}^t\right|$ indicates the number of filters of $\widetilde{W}^t$. More specifically, as shown in Fig. 1(a), GT-filters Alg dynamically expand the filters from an initial small network into a reasonable large one. Starting from a small number of filters (e.g. 2) of conv layer, more and more filters tend to be with non-zero values as the

algorithm iterates. Every $J$ epochs, we can compute the ratio of Eq. (9): if this ratio passes a pre-set threshold $\tau$, we add the same number of new filters as existing filters[1] into $W$; otherwise, we will not grow any filter in this epoch. Then we continue optimizing all the weights from training data; this process is repeated until the loss does not change much or maximum epochs is reached.

**Remarks**. We highlight several insights of our GT-filter Alg (1) As a trivial case, our GT-filters Alg can be directly utilized to boost neurons in fully connected layer. (2) GT-filter Alg can be implemented in parallel to boost each individual layer simultaneously.

### 3.3 GROWING AND TRAINING LAYERS ALGORITHM (GT-LAYERS ALG )

The GT-filters Alg is designed to dynamically add and train filters in one conv layer, rather than adding new conv layer of the whole network. To overcome this limitation, we further propose the GT-layers Alg, which can learn to boost layers of one network. We assume the network (e.g, VGG, or ResNet) is composed of several blocks (e.g. VGG block or Residual blocks) with necessary transition layers (e.g., pooling layer) between two blocks; each block has many conv layer of the same size of conv filters. The number of total blocks is fixed in the network; and only layers are boosted in GT-layers Alg.

The GT-layers Alg has two key steps: (1) learning the filter configuration of each conv of each block; and (2) boosting layers of each block. Specifically, in Step (1): given an initial network of $B$ blocks; and each block has only one conv layer (for plain net) or one BasicBlock (for ResNet) in He et al. (2016a) which has 2 conv layers; we apply GT-filters Alg to boost filters of each conv layer of each block one by one. GT-filters Alg will find the final number of filters of each conv layer as $M_i$ $(i = 1, \cdots B)$. In Step (2): we initialize a network which has each block of one conv layer (for plain net) or one BasicBlock (for ResNet) in He et al. (2016a) which has 2 conv layers, each layer or layers of BasicBlock consisting of $M_i$ $(i = 1, \cdots B)$ filters. We train the network from the scratch by boosting the layers from bottom blocks to up blocks of networks, following the data streaming from input to output, in Fig. 1 (b).

Along the training path, if the Eq. (9) is established, and this ratio passes the threshold $\tau$ for the block $b$, we denote the training accuracy as $Acc_{before}$; and further add another conv layer or BasicBlock of $M_b$ filters in each layer to this block, and each filter will be initialized as He et al. (2015), with zeros initialization for the corresponding dimension in $Z$ and $\Gamma$; We continue the training process for $J$ epochs, and denote the training accuracy as $Acc_{after}$. If

$$|Acc_{after} - Acc_{before}| < \epsilon, \tag{10}$$

this indicates that block $b$ has enough capacity, and we will not add layers or filters for block $b$. We continue the training process until model converged or maximum budget (epochs) is reached.

**Remarks**. We have several reasonable simplification in GT-layers Alg. (1) The filters of each conv layer in the same block should be the same, since it is a standard practice in the most state-of-the-art manually designed structures, *e.g.,* VGG and ResNet family. (2) We still utilize Eq. (9) as the metric to control the capacity of networks. Critically, by introducing the sparse set $\Gamma$, the learned model is still over-parameterized in general, and yet with controllable total parameters. Thus the boosted network can enjoy the best of two worlds. (3) We have to boost layers from bottom to up blocks of the networks, since we rely on Eq. (10) to judge whether to stop boosting layers for each block.

## 4 EXPERIMENTS

**Dataset and Implementation.** We conduct the experiments to evaluate our algorithms on MNIST, and CIFAR10/100 datasets. Unless otherwise specified, the hyper-parameters of Split LBI are $\kappa = 1$, $\nu = 100$, $\alpha = 0.01$, with batch size 128. To validate GT-filter Alg the initial network used has 20 filters for each conv layer, and 100 neuron in each FC layer by default. For GT-layers Alg, the initial VGG-like network has one input conv layer, and 4 blocks, with 1 conv layer of 10 filters; and the initial ResNet-like network for GT-layers Alg, has one input conv layer, and 4 blocks, each block has 1 BasicBlock in He et al. (2016a), and each BasicBlock with 2 conv layers of 20 filters in each layer. We set the hyper-parameters as $J = 40$, $\epsilon = 0.3$, and $\tau = 0.4$ by default. After finishing

---

[1]$Z$, and $\Gamma$ will add corresponding dimensions, initialized as zeros; and the newly added parameters of $W$ are randomly initialized as He et al. (2015).

| | filter number of each conv layer @ filter-size | | Accuracy(%) | | |
|---|---|---|---|---|---|
| Dataset | Seed Net | Boosted Net | LB | UB | GT-filters Alg |
| MNIST | **[2@3*3]** | **[8@3*3]** | 96.11 | 98.61 | 98.61 |
| CIFAR10 | **[2@3*3]** | **[8@3*3]** | 46.11 | 63.45 | 63.21 |
| | 8@5*5-**[2@3*3]** | 8@5*5-**[172@3*3]** | 51.95 | 80.90 | 79.76 |
| | 8@5*5-8@3*3-**[2@3*3]** | 8@5*5-8@3*3-**[190@3*3]** | 57.21 | 80.69 | 80.31 |
| CIFAR100 | 8@5*5-**[2@3*3]** | 8@5*5-**[154@3*3]** | 22.37 | 52.99 | 52.36 |
| | 8@5*5-8@3*3-**[2@3*3]** | 8@5*5-8@3*3-**[180@3*3]** | 23.00 | 53.19 | 52.87 |

Table 1: Results of boosting filters (GT-filters) of a particular one layer network on MNIST, Cifar-10, and Cifar-100. [-] indicates the corresponding layer boosted. ( Here 8@3*3 means that this layer has 8 filters with size of 3 *3)

adding filters/layers, we decrease the learning rate by 1/10, continue training 70 epochs; and then further decrease the learning rate by 1/10 again, and go on training 30 epochs.

## 4.1 EXPERIMENTS ON GROWING FILTERS BY GT-FILTERS ALGORITHM

**Boosting Shallow Networks.** We explore the performance that GT-filters Algorithm boosts one conv layer shallow networks to much wider ones on MNIST and CIFAR10/100 datasets. Given a network of initially a small number of filters (denoted as Seed Net), our GT-filters Alg will add and train filters to produce a network of large number of filters (denoted as Boosted Net). Here, we introduce two competitors: (1) Lower Bound (LB): directly training Seed Net by SplitLBI from the scratch; (2) Upper Bound (UB): directly training a network having the same structure as Boosted Net by SplitLBI from the scratch. Essentially, LB and UB serves as the lower and upper bound performance for the network learned by our GT-filter Algorithm. All models are trained by SplitLBI in 1000 epochs. We report the network structure and results in Tab. 1. Our GT-filters Alg boosts the filters of the network, denoted as Boosted-Net, which performs almost the same as UB and much better than LB of all cases in Tab. 1, this indicates that our GT-filters Alg indeed successfully boosts the filters of networks.

**Boosting Deep Networks.** We further explore our GT-filters Alg boosting deep neural networks with more filters on CIFAR10 dataset. We employ the VGG and ResNet families as the backbone, since they are most typical models of plain net and skip-connection net. Our algorithm is compared against several naive ways in gradually boosting the filters in Tab. 2. (1) Random-layers-adding-filters (Random): After training very $J$ epochs, we randomly select half number of all layers, double the filters of these selected layers, initialize the newly added filters, and continue to training by SGD. Repeat these steps until meet stopping condition (2) Ordering-layers-adding-filters (Order): We equally divide all layers in bottom and upper layer groups. After training every $J$ epochs, we double the filters of each layer in each group in turn and go on training by SGD. Repeat these steps until meeting the stopping condition.

We set $J = 30$ for, and $\tau = 0.5$ for our GT-filters Alg. For competitors, we adopt the stopping growing filters policy that, aftter growing filters and training $J$ epochs by referring to Eq. (10), the increased validation is less than $1\%$. The maximum training epochs is set as 300.

Figure 2 shows the growing and training process. In general, the training process of Random-layers-adding-filters, Ordering-layers-adding-filters and our GT-Filters Alg are very close to each other. At the first time of growing filters, the performance of networks sharply decreased as along as adding filters, partly due to the fact that the initialization of networks after adding filters are far from any optima. The results are given in Table 2. The two baselines and our GT-Layer Alg achieved nearly the same accuracy, with orders of magnitude higher model size than ours. Interestingly, our GT-filters Alg found a sparse network with small number of filters in each layer and with only $1/7$ parameters comparing to the two baselines. This experiment suggests that our GT-Layers Alg indeed could boost filters for deep networks.

**Ablation study of GT-filters Alg.** We conduct the ablation study and validate the efficacy of different hyper-parameters of $J$ and $\tau$ in Tab. 3 and Tab. 4, trained on CIFAR10 dataset. Our model is compared against VGG-16 network trained by 350 epochs. We found the higher $J$ value, the larger

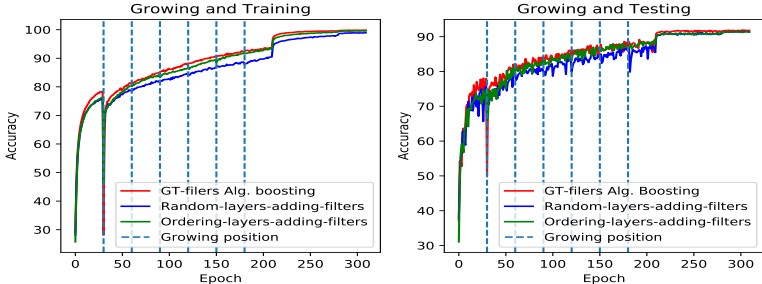

Figure 2: Different growing filters strategies: GT-filters Alg, Random, Order. Left subfigure is the training process, and right is corresponding testing accuracy. After growing finished, there will be a learning rate decay and go on training for some epochs.

| Method | Boosted-Net of each method. | Params | Acc(%) |
|---|---|---|---|
| Random | 256, [128], [128, 64], [128, 128, 256], [512, 128, 256], [256, 128, 256], [2048, 1024] | 6.92 M | 91.45 |
| Order | 256, [256], [256, 256], [256, 256, 256], [256, 128, 128], [128, 128, 128], [1024, 1024] | 6.22 M | 91.40 |
| GT-filters Alg | 40, [80], [160, 160], [160, 80, 80], [80, 40, 20], [20, 20, 20], [100, 100] | 0.90M | 91.82 |

Table 2: Results of boosting filters (GT-filters) of VGG16 networks. [-] indicated the convolution filters of corresponding layer, the last [-] indicates the number of linear units.

boosted network with better performance. Besides, all of our boosted networks are in a low level of parameter number, from $1/30$ to $1/7$ comparing to VGG-16 network, but with high performance. Especially, when $J = 50$, our final boosted networks have comparable to VGG-16. We argue that these results are reasonable, since the $\Gamma$ set may not be well trained with smaller $J$, e.g., $J = 20$, and Eq. (9) may not be met. In Tab. 4, we set $J = 40$, our boosted models using $\tau = 0.3$, 0.4 and 0.5 achieved high performance as well as low level of parameter number. Our hyper-parameter $\tau = 0.5$ can result in the boosted network that have slightly inferior performace, and yet much less parameters than VGG-16. We highlight that our boosted results are not sensitive to different values of $\tau$. Overall, this experiment suggests the efficacy of our GT-filters Alg in boosting filters.

**Growing filters in ResNet-18.** It is interesting to investigate whether we can boost a ResNet network. Here we start with ResNet18, given the same number of layers and structure as the Standard ResNet-18 (S-ResNet-18). We conduct the experiments on CIFAR10/100 datasets. Our GT-filters Alg generates the Boosted ResNet-18 (Boosted-Net). In addition, we also introduce the Total FLOPs Budge ResNet-18 (TFB-ResNet-18) which is trained standard ResNet-18 by the same total FLOating Point operations (FLOPs) as our B-ResNet-18. To train TFB-ResNet-18, we set $\alpha = 0.05$, 0.01, and 0.001 after every 1/3 total FLOPs. The results are shown in Tab. 5. We find that on CIFAR10, our B-ResNet-18 has much less parameters, about $1/5$ comparing to S-ResNet-18, but achieve comparable performance to S-ResNet-18 and even higher than TFB-ResNet-18. On CIFAR100, our boosted network performed a little worse than S-ResNet-18 but we performed much better than TFB-ResNet18 and we use less than $2/3$ number of parameters comparing to S-ResNet-18. The most important is that total FLOPs we used in boosting are not enough for training a ResNet18 network, which demonstrates the effectiveness of our algorithms.

| Epoch $J$ | Boosted-Net by GT-filter Alg. | Params | Accuracy(%) |
|---|---|---|---|
| 20 | 20, [20], [20, 20], [20, 20, 20], [20, 20, 20], [20, 20, 20] | 0.06M | 83.10 |
| 30 | 40, [80], [160,160], [160, 80, 80], [80, 40, 20], [20, 20, 20] | 0.90M | 91.80 |
| 40 | 40, [80], [160,160], [160,160,160], [80,80,40], [40, 40, 40] | 1.33M | 91.40 |
| 50 | 80, [160], [320, 320], [320, 320, 320], [160, 80, 40],[40, 40, 40] | 4.93M | 92.80 |
| VGG-16 | 64, [64], [128, 128], [256, 256, 256], [512, 512, 512], [512, 512, 512] | 33.65M | 92.64 |

Table 3: Varying Epoch $J$ in boosting VGG family network on CIFAR10 dataset by GT-filters Alg. ($\tau = 0.5$). [20,20] indicates a block of two conv layer; each has 20 filters. Each network has an input conv layer.

| $\tau$ | Found Filters of Each Layer | Params | Accuracy(%) |
|---|---|---|---|
| 0.3 | 80, [160], [320, 320], [320, 320, 320], [160, 80, 80], [40, 80, 80] | 5.04M | 92.55 |
| 0.4 | 80, [160], [160, 320], [320, 320, 320], [160, 80, 80], [40, 40, 40] | 4.28M | 92.70 |
| 0.5 | 40, [80], [160, 160], [160, 160, 160], [80, 80, 40], [40, 40, 40] | 1.33M | 91.40 |
| VGG-16 | 64, [64], [128, 128], [256, 256, 256], [512, 512, 512], [512, 512, 512] | 33.65M | 92.64 |

Table 4: Varying threshold $\tau$ in boosting VGG family network on CIFAR10 dataset by GT-filters Alg ($J = 40$). [160, 160] indicates a block of two conv layer; each has 160 filters. Each network has an input conv layer.

|  | Model | Params | Total FLOPs | Accuracy(%) |
|---|---|---|---|---|
| CIFAR10 | Boosted-Net | 2.30M | 560.93G | 93.41 |
|  | S-ResNet-18 | 11.74M | 4091.50G | 93.02 |
|  | TFB-ResNet-18 | 11.74M | 561.12G | 91.2 |
| CIFAR100 | Boosted-Net | 7.38M | 825.07G | 73.46 |
|  | S-ResNet-18 | 11.74M | 2338.00G | 75.5 |
|  | TFB-ResNet-18 | 11.74M | 829.99G | 67.0 |

Table 5: Under the same budget of computational cost (total FLOPSs), our growing filter strategy for boosting network achieves better accuracy and smaller complexity than standard training networks, comparable accuracy to those with much more cost.

## 4.2 EXTENDING TO GROWING LAYERS BY GT-LAYERS ALGORITHM

Section 4.1 explores our GT-filers Alg and shows that our method achieved good results in boosting filter for fixed deep networks. In this section, we study our GT-layers Alg in boosting both filters and layers for a shallow 'seed' network on CIFAR10 and CIFAR100 datasets. Here we use the initial plain net, and initial residual net referring to VGG net and ResNet, individually. The structures are: (1) **(ResNet)** the same architecture as ResNet in He et al. (2016b): it has 4 blocks, and each block has several BasicBlocks and 2 convolutional layers in each BasicBlock. We initialize each convolutional layer with 20 filters. (2) **(PlainNet)** a VGG-like plain net: it has 4 blocks divided by pooling layers, and each block has several conv layers with 10 filters in each conv layer. The processes of GT-layer Alg. have two parts: firstly we grow filters for seed net and get the configuration of filter number of all blocks, then we start growing layers. Note that we keep same filter number of conv layers inside of a block, so we first search the filter number configuration.

We also compare two types of DNNs of Autogrow (Wen et al., 2019): (1) **(Basic4ResNet)** a variant of ResNet with basic residual blocks 3 used for ImageNet in He et al. (2016b); (2) **(Plain4Net)** a VggNet-like plain net by removing shortcuts in Basic4ResNet.

Table 6 compares the growing results of Autogrow and our boosting results using GT-layers Alg. Autogrow is one of the most efficiency methods in growing layers of networks. Specifically, Autogrow can grow layers from a seed network, but their approach does not explore the filter configuration of each block. If compared aganist our GT-layers Alg, the results networks have much deeper with a large number of parameters. On CIFAR10 dataset, the Boosted-Net by our GT-layers Alg performs as good as Plain4Net and Basic4Net models by Autogrow. However, our boosted networks are much shallower than the found nets of Autogrow but having nearly the same performance. For

| Dataset | Method | Net Type | Layers | #Param | Acc(%) |
|---|---|---|---|---|---|
| CIFAR10 | Autogrow | Basic4ResNet | 178 | 136.51M | 95.49 |
|  |  | Plain4Net | 138 | 105.06M | 94.20 |
|  | Ours | ResNet($\tau$=0.4,J=40,$\epsilon$=0.3) | 26 | 15.33M | 95.11 |
|  |  | PlainNet($\tau$=0.5,J=30,$\epsilon$=0.3) | 16 | 2.58M | 94.25 |
| CIFAR100 | Autogrow | Basic4ResNet | 202 | 109.38M | 79.47 |
|  |  | Plain4Net | 124 | 92.59M | 73.91 |
|  | Ours | ResNet($\tau$=0.4,J=30,$\epsilon$=0.3) | 24 | 31.33M | 75.70 |
|  |  | PlainNet($\tau$=0.5,J=30,$\epsilon$=0.3) | 15 | 13.33M | 75.35 |

Table 6: Comparison of Autogrow and our algorithm in found nets and performance, where ours achieves comparable accuracies with much smaller model complexity.

| $\epsilon$ | Found Net | Layers | Accuracy(%) |
|---|---|---|---|
| ResNet18 | [2, 2, 2, 2] | 18 | 93.02 |
| ResNet34 | [3, 4, 6, 3] | 34 | 94.10 |
| 0.1 | [5, 2, 4, 3] | 30 | 94.60 |
| 0.2 | [3, 2, 4, 2] | 24 | 94.80 |
| 0.3 | [5, 3, 2, 2] | 26 | 95.10 |

Table 7: Comparison between the standard ResNets and our boosting algorithm at different thresholds ($\epsilon$) with $J = 40$ and $\tau = 0.4$ on CIFAR10 dataset. The boosted networks have stable performance with varying thresholds $\epsilon$ and improve the standard ResNet34 with less number of layers.

example, on CIFAR10, our GT-layers Alg found a 16 layer VGG-like network with 2.58M parameters, Autogrow found a 138 layer network with approximate 105.06M parameters. On CIAFR100, by using plain net, our algorithm not only boosts much shallower networks and small number of parameters, but also performs much better than the models found by Autogrow. In general, our GT-Layer Alg could not only efficiently boost networks from shallow to properly deep, but also achieve very good performance.

We also conduct ablation study of the hyper-parameter $\epsilon$. We compare the results of different $\epsilon$, and the results of standard ResNet18 and ResNet34 trained for 350 and 300 epochs, respectively. Table 7 shows the boosting results of different $\epsilon$ and standard models. As expected, smaller $\epsilon$ will find a deeper network. The accuracy of boosted models using different $\epsilon$ is not so much difference from each other. This indicates that our GT-Layer Alg is not sensitive to small $\epsilon$. Besides, All of our found networks performed equally or better comparing to standard networks. This suggest that our algorithm could have very good performance in boosting layers.

## 5 CONCLUSION

In this paper, we study the novel task of boosting network and propose an approach that simultaneously growing and training filters and layers: GT-filters Alg and GT-layers Alg. With experiments on VGG and ResNets, these algorithms could efficiently boost fixed networks from a small number of filters in each layer and boost shallow seed networks, respectively, with comparable accuracies to big models but remarkably economic representations.

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
