# OpenReview forum: "Boosting Network: Learn by Growing Filters and Layers via SplitLBI"
_ICLR.cc/2020/Conference — Reject_

### Official Review · AnonReviewer1 · 2019-10-23
**Official Blind Review #1**

**Rating:** 3

**Review:**

Summary:
This paper focuses on topic of searching for the optimal architecture for the deep network. Building on the split linearized bregman iteration strategy, the authors propose two practical algorithms to boost network, namely GT-filters Alg and GT-layers Alg. The proposed algorithms can simultaneously grow and train a network by progressively adding both convolutional filters and layers. The experiments conducted on VGG and ResNets display the comparable accuracies between the BoN and the standard big models, but with much more compact representations and balanced computational cost.

Strengths:
1 The authors introduce two simple but practical algorithms for augmenting the architectures of deep network. The quite promising results are achieved on baselines, w.r.t. the balance of prediction performance and model complexity with the budgeted computational resources.
2 The paper is clearly written and easy to follow.

Weaknesses:
1 The proposed BoN is built upon the existing SplitLBI algorithm that can identify the sparse approximation of the weight structure. The contributions are mainly from incorporating the adaptive criteria for adding filters and layers. Thus contribution is incremental and novelty is limited.
2 The projection operator in equation (4) is a key step of the boosting procedure, but is not clearly defined and explained.
3 For experiments, results shows good results on simple baselines, more complicated or large-scale datasets should be included for evaluations. Also, only Autogrow is compared to the proposed method, to make the results more convincing, other architecture searching approaches (e.g. NAS) should be added to comparisons.

Other questions:
1 For the GT-layers Alg, would it more efficient to boost layer first before boosting the filters?
2 Will GT-layers be robust to the case when the number of filters is overly specified?

**Experience Assessment:**

I have published one or two papers in this area.

**Review Assessment: Checking Correctness Of Derivations And Theory:**

I did not assess the derivations or theory.

**Review Assessment: Checking Correctness Of Experiments:**

I assessed the sensibility of the experiments.

**Review Assessment: Thoroughness In Paper Reading:**

I made a quick assessment of this paper.

---

> ### Author Response · Authors · 2019-11-14
> **For Reviewer #1**
>
> Very thanks very much for the suggestions and comments. We answer the specific questions below,
>
> Q1 : The proposed BoN is built upon the existing SplitLBI algorithm that can identify the sparse approximation of the weight structure. The contributions are mainly from incorporating the adaptive criteria for adding filters and layers. Thus contribution is incremental and novelty is limited.
> A1 : Thanks. Actually not. Despite our algorithm is built upon SplitLBI, which is one of most recent optimizers, the main focus of this paper is about boosting network, i.e., efficiently growing filters and layers of network. The goal of our paper is boosting networks, and we propose new algorithm to reveal our goal by using Split LBI as a tool. The original SplitLBI is an optimization algorithm and the usage of \Gamma is for sparsity.
>
> We summarize the contributions: (1) A novel learning paradigm - Boosting net- work (BoN), is for the first time, studied in this paper; and one starts from simple models, delving into complex trained models progressively. (2) We propose a novel GT-filters Alg, which can si- multaneously effectively grow the filters of each layer, and train the network parameters, given an initial network. (3) We present GT-layers Alg which grows and trains layers, by exploring the over- parameterized model weight and structural sparsity model weight set.
>
>
> Q2 : The projection operator in equation (4) is a key step of the boosting procedure, but is not clearly defined and explained.
> A2 :  Thanks. We add some explanations.
> “This above equation means $(W^{t}$ is projected on  $\Gamma^{t}$, and
> the selected subset
>  $\widetilde{W}^{t}$ include the parameters existed in both $W^{t}$ and  $\Gamma^{t}$. “
>
> Q3 : For experiments, results shows good results on simple baselines, more complicated or large-scale datasets should be included for evaluations. Also, only Autogrow is compared to the proposed method, to make the results more convincing, other architecture searching approaches (e.g. NAS) should be added to comparisons.
>
> A3: Thanks. But in our paper, we demand the following properties of an algorithm qualified as BoN:
> (1)It should incorporate both architecture growth (including filters and layers) and parameter learning simultaneously, in which the width and depth of network can be gradually updated, and the parameters of network should be updated at the same time;
> (2)It should provide a comparable classifier for prediction tasks, as the state-of-the-art hand- crafted architectures on the same dataset;
> (3)Its computational requirements, the total parameters of final boosted network, and memory footprint should remain bounded, ideally in the same order of magnitude as training a manually engineered architecture on the same dataset.
>
> The first two criteria express the essence of boosting network; the third criterion identifies the key difference from NAS and other trivial or brute-force solutions, such as randomly searching.
> As the field of representation learning moves closer towards artificial intelligence, it becomes important to efficiently and simultaneously learn both the structures and parameters of a network from arbitrary classes on mobile devices or even Internet of Things (IoT) devices.
> To sum up, the goal of our work is boosting networks in an efficient way, which is quite different from NAS, which is searching for good structure. It’s very hard for our algorithm to make a direct and far comparison to NAS, in term of GPU computational cost, and total parameters of final network structures.
>
>
> Q4 : For the GT-layers Alg, would it more efficient to boost layer first before boosting the filters?
> A4 : Thanks. Thanks for this point. The GT-Filters Alg in GT-Layers algorithm aims at finding the optimal configuration of the filter number.  Generally and practically, it is a good practice to use the same number of filters in each convolutional layer, as has done in many previous manually designed architectures. On the other hand, some of our pilot experiments suggest that, if alternatively growing filters and layers, it may greatly improve the computational cost, but do not incur significant performance improvement.  Thus, the algorithm is designed as boosting filters, and then boosting layers.
>
>
> Q5: Will GT-layers be robust to the case when the number of filters is overly specified?
> A5: Thanks.  If the number of filters overly specified, the model may be also over-parameterized.

---

### Official Review · AnonReviewer2 · 2019-10-26
**Official Blind Review #2**

**Rating:** 6

**Review:**

This paper proposes an architecture search method for deep convolutional neural network models that progressively increases the number of filters per layer as well as the number of layers, and the authors refer to this general approach as boosting networks. The algorithm for increasing the number of filters is based on split linear Bregman iteration, and the algorithm for increasing the number of layers proceeds block by block, increasing the layers per block until the accuracy does not increase. The experiments convincingly demonstrate gains in performance and smaller network sizes compared to baseline models, naive boosting methods, and a related method called Autogrow.

In my view, there are two main areas for further improvement for this work. First, the GT-Layers algorithm can be better motivated. Why is GT-Filters only run once at the beginning, rather than iteratively as the number of layers increases? Why does the procedure go from bottom blocks to up blocks (and, by the way, what are bottom blocks and up blocks)? Why measure training accuracy to determine when to add layers? I understand that these are all fairly heuristic choices, but nevertheless there needs to be proper motivation for all of the above.

Second, an additional effort should be made to compare to additional prior work in architecture search. It seems like the authors are suggesting that this method should be more computationally efficient and find smaller architectures, and demonstrating this empirically would greatly strengthen the paper. In particular, the existing results depicting the final number of parameters in the learned model are particularly striking to me. I appreciate that the authors included an experiment showing that a standard ResNet cannot be trained with the same number of FLOPs as the network found by your method. Can a similar analysis be made for wall clock time, i.e., how long the models actually take to train? A similar study (FLOPs, wall clock time, etc.) would also be very useful for the current comparison to Autogrow, as these metrics are often just as important, if not more important, than the number of parameters of the final model.

A thorough pass through the paper for spelling and grammar would be very useful.

**Experience Assessment:**

I do not know much about this area.

**Review Assessment: Checking Correctness Of Derivations And Theory:**

N/A

**Review Assessment: Checking Correctness Of Experiments:**

I assessed the sensibility of the experiments.

**Review Assessment: Thoroughness In Paper Reading:**

I read the paper at least twice and used my best judgement in assessing the paper.

---

> ### Author Response · Authors · 2019-11-14
> **For Reviewer #2**
>
> Thanks very much for the suggestions and comments. We answer the specific questions below,
>
> Q1: The GT-Layers algorithm can be better motivated. Why is GT-Filters only run once at the beginning, rather than iteratively as the number of layers increases?
> A1 : Thanks for this point. The GT-Filters Alg in GT-Layers algorithm aims at finding the optimal configuration of the filter number.  Generally and practically, it is a good practice to use the same number of filters in each convolutional layer, as has done in many previous manually designed architectures. On the other hand, some of our pilot experiments suggest that, if alternatively growing filters and layers, it may greatly improve the computational cost, but do not incur significant performance improvement.
>
>
> Q2 : Why does the procedure go from bottom blocks to up blocks and, by the way, what are bottom blocks and up blocks?
> A2 : Following the data streaming from input to output, we boosting the layers from bottom blocks to up blocks as in Fig. 1(b).
> This idea is rooted in Deep Belief Network, which utilized unsupervised pre-training layer-by-layer, given the input data streaming.
>
> Q3: Why measure training accuracy to determine when to add layers?
> A3 : Thanks. Yes, in our Boosting algorithms,  one starts from simple models, delving into complex trained models progressively. If the training accuracy is not high, it indicates that the network doesnot have enough capacity (not over-parameterized) to learn the training data. Thus it is a good measure.
>
>
> Q4 : Can a similar analysis be made for wall clock time, i.e., how long the models actually take to train? A similar study (FLOPs, wall clock time, etc.) would also be very useful for the current comparison to Autogrow, as these metrics are often just as important, if not more important, than the number of parameters of the final model.
>
> A4: The running time given in the paper of Autogrow is the experiments on ImageNet, in order to compare the running time on CIFAR10, we run the source code of Autogrow released by the author and running on our single TITAN X (Pascal) GPU. Results are:
>
> Autogrow on Cifar10 (using ResNet block):
> main_gradual.py :  ACC- 93.75%; running-time-9.74h;
> main_add.py:         ACC- 92.57%; running-time-7.36h
>
> Our method on Cifar10:
> ResNet:     ACC-94.60% ; time-4.0h;
> (PlainNet: ACC-94.65%; time-3.0h)

---

### Official Review · AnonReviewer3 · 2019-10-27
**Official Blind Review #3**

**Rating:** 3

**Review:**

This paper studies a very interesting topic: automatically grow filters and layers in neural networks and find an "optimal" width and depth for neural networks. The method is motivated by SPLITLBI, and its effectiveness is verified by experiments and comparison with AutoGrow. I tend to accept this paper, before the following questions can be answered:
1. I guess the major issue in this paper is that the method is not clearly explained and rigorously formulated, although it's an extension of SPLITLBI.
-- 1.1. what's Γ? Is it a copy of W or not? What's the exact mathematical function of loss L() w.r.t. W and Γ? How does the neural architecture change after adding Γ?
-- 1.2. why Γ can be approximated by gradients in Eq. (3)? What's the intuition behind?
-- 1.3. why Γ is necessary? How does it compare with enforcing group Lasso on W directly, like what was done in Nonparametric Neural Networks [1]?
Without clarifying those, people can hardly learn from and use this paper.

2. Experiments
-- 2.1. Include the learned width in Table 1.
-- 2.2. In comparison with AutoGrow, the pairs of ResNet is fair, but the pairs of PlainNet is hard to judge because different neural architectures are used. AutoGrow uses 4 blocks while this paper uses 5 blocks. It's unclear if the benefit comes from the method or just from an additional block.
-- 2.3. In the experiments of layer growing, please clarify if filter growth is also applied or not.
-- 2.4 clarify "their growing process is not efficient." If I read the AutoGrow paper correctly, efficiency is one of the their claims and they showed that the growing process is as fast as "training a single DNN", and they scaled to ImageNet, which is not covered in this paper.

Minors:
1. networks with "20 filters" and "100 neurons" are used as the seeds. How critical are they?
2. "To the best of our knowledge, this is the first algorithm for BoN that can simultaneously learn the network structures and parameters from training data." is over-claimed. Lots of pruning methods can do it, although they start from a large one and prune it down.

[1] Philipp, George, and Jaime G. Carbonell. "Nonparametric neural networks." arXiv preprint arXiv:1712.05440 (2017).

**Experience Assessment:**

I have published one or two papers in this area.

**Review Assessment: Checking Correctness Of Derivations And Theory:**

N/A

**Review Assessment: Checking Correctness Of Experiments:**

I carefully checked the experiments.

**Review Assessment: Thoroughness In Paper Reading:**

I read the paper at least twice and used my best judgement in assessing the paper.

---

> ### Author Response · Authors · 2019-11-14
> **For reviewer #3**
>
> Thank you very much for your comments.
>
> Q1: The method is not clearly explained and rigorously formulated.
>
> (Q1-1) what's Γ? Is it a copy of W or not? What's the exact mathematical function of loss L() w.r.t. W and Γ? How does the neural architecture change after adding Γ?
> (A-1): Yes, we had clearly defined \Gamma in Eq (3). \Gamma is the structural sparsity parameter to explore important subnetwork architectures by inverse scale space where those important parameters become nonzero faster than others. Mathematically, in Eq (3), \Gamma learned to approximate W but not a copy of W. Thus the \Gamma is initialized as the same dimension as W, while most values of \Gamma should be zeros (sparse). Here W is the model parameters of deep networks.
>
> We update the loss function w.r.t W and Γ in the revised version
>
>
> (Q1-2) why Γ can be approximated by gradients in Eq. (3)...
> (A-2). Thanks. We update this point, and give more detailed mathematical definition, and intuition in the revised
>
>
> (Q-3) why Γ is necessary? How does it compare with enforcing group Lasso on W directly, like what was done in Nonparametric Neural Networks [1]?
> (A-3). To efficiently boosting the network, we need the merit of learning both an over-parameterized model weight set (Over-Par set) as the Stochastic Gradient Descent (SGD), and structural sparsity model weight set (Stru-Spa set) in a coupled inverse scale space. Thus the weight parameter W is for Over-Par set, and Γ is for structural sparsity set. Thus it is necessary to introduce Γ.
>
> One can enforce group lasso on W or as Nonparametric Neural Networks [1], but we still need the structural sparsity set Γ, to help measure whether the filters or layers should be boosted. In that sense, the structural sparsity set selected by  Nonparametric Neural Networks, can also potentially be utilized as an alternative to Γ; and we take it as a future work.
>
>
> Q2: Improvement on experiments.
>
> (Q2-1) Include the learned width in Table 1.
> (A2-1): Thanks. The learned width is actually shown in Tab 1. Particularly, in Tab 1 of our unrevised paper, the “UP-Net” column describes our learned width of each network, for example, 8@3*3 means kernel size is 3 with 8 channels we learned. We highlight this point in the revised version.
>
> (Q2-2) the pairs of PlainNet is hard to judge because different neural architectures are used. AutoGrow uses 4 blocks while this paper uses 5 blocks. ...
> (A2-2). Thanks. For fair comparison we add one experiment on 4-blocks plainnet.  It shows that our method has much less parameters while still keeps almost same performance.
>
> 4-Block-PlainNet(J=30,\epsilon=0.3,\tau=0.5):
> CIFAR10:  ACC-94.30%; Layer-16; Param-2.577M;
> CIFAR100:  ACC-75.35%; Layer-15; Param-13.332 M;
>
> Compared with 5-Block-PlainNet, which has the results as
> CIFAR10:  ACC-94.65； Layer-19; Param-7.70M;
> CIFAR100:  ACC-75.66； Layer-18; Param-29.83M;
>
>
> (Q2-3) In the experiments of layer growing, please clarify if filter growth is also applied or not.
> (A2-3). In the experiments of layer growing, we do use filter growth, as explicitly explained in Sec. 3.3 The detailed procedure is : 1). Initialize the base network. 2). Grow filters to determine appropriate channels of each block(simply, layers between 2 pooling layers in PlainNet called a block; and layers having same output size in ResNet called a block), i.e., “learning the filter configuration of each conv of each block;”  3). After getting number of channels, we begin to grow layer.  We further highlight this point.
>
> (Q2-4) clarify "their growing process is not efficient." AutoGrow is very efficiency ...
> (A2-4).  Yes, indeed, AutoGrow is one of the most efficiency methods for the proposed BoN task; but it is less efficient than ours. we elaborate it in the revised paper,
>
> We also empirically compare the running time:
> We run the source code released by the author with their default setting, and running on our single TITAN X (Pascal) GPU.
> Autogrow on Cifar10 (using ResNet block):
> main_gradual.py : ACC- 93.75%; running-time-9.74h;
> main_add.py: ACC- 92.57%; running-time-7.36h
> Our method on Cifar10:
> ResNet: ACC-94.60% ; time-4.0h;
> (PlainNet: ACC-94.65%; time-3.0h)
>
> Q3: Some minor issues:
>
> (Q3-1) networks with "20 filters" and "100 neurons" are used as the seeds. How critical are they?
> (A3-1).For the initial number of units in linear layer, we have used other trial such as 50, 200, 1000 neurons, the final results do not have significant difference: the accuracy difference is <0.02%. And for the initial number of filters, we also tried 8, 10, 16 filters, the results also do not have significant difference. This is to say, the initial seed is not so critical, so we stick to 20 filters, and 100 neurons in our experiments for convenience.
>
> (Q3-2) "To the best of our knowledge, .. over-claimed.
> (A3-2).Thank. We revised the claim as suggested.
>
>
> [1] Huang et al., Split LBI: An iterative regularization path with structural sparsity. NeurIPS 2016.

---

### Author Response · Authors · 2019-11-14
**Summary of Changes**

We thank all the reviewers for their insightful and constructive comments. We have substantially revised the paper as suggested by the reviewers, and summarize the major changes as follows:

1. In Introduction, we explicitly highlight our contributions, and explain that (1) we for the first time, define the task of boosting network, and present the algorithms of boosting network. In particular,  our algorithm is built upon the existing deep network optimizer -- SplitLBI,

2. In methodology,  we rewrote and clearly gave more mathematical definition, and intuitions about the background of SplitLBI in Sec. 3.1, as suggested. Specifically, the proximal map, and loss functions of SplitLBI have been more clearly defined.

3. In section 3.2, we give the intuition about Eq (8) (old Eq (4)):
    “This above equation means $(W^{t}$ is projected on $\Gamma^{t}$, and the selected subset  $\widetilde{W}^{t}$ include the parameters existed in both $W^{t}$ and $\Gamma^{t}$. “

4. As suggested by reviewers, we add some explanations to Autogrow: “Autogrow is one of the most efficiency methods in growing networks. Specifically, Autogrow can grow layers from a seed network, but their approach does not explore the filter configuration of each block. If compared aganist our GT-layers Alg, the results networks have much deeper with a large number of parameters; and thus their growing process is less efficiency than ours.”

5. We also update the experimental PlainNet results in Tab. 6, by using 4 blocks as the initialization, as suggested.

---

### Decision · Program_Chairs · 2019-12-19

**Decision:**

Reject

**Comment:**

This paper considers how to learn the structure of deep network by beginning with a simple network and then progressively adding layers and filters as needed. The paper received three reviews by expert working in this area. R1 recommends Weak Reject due to concerns about novelty, degree of contribution, clarity of technical exposition, and experiments. R2 recommends Weak Accept and has some specific suggestions and questions. R3 recommends Weak Reject, also citing concerns with experiments and writing. The authors submitted a response that addressed many of these comments, but R1 and R3 continue to have concerns about contribution and the experiments, while R2 maintains their Weak Accept rating. Given the split decision, the AC also read the paper. While we believe the paper has significant merit, we agree with R1 and R3 on the need for additional experimentation, and believe another round of peer review would help clarify the writing and contribution. We hope the reviewer comments will hep authors prepare a revision for a future venue.